# Longitudinal Humoral and Cellular Immune Responses Following SARS-CoV-2 Vaccination in Patients with Myeloid and Lymphoid Neoplasms Compared to a Reference Cohort: Results of a Prospective Trial of the East German Study Group for Hematology and Oncology (OSHO)

**DOI:** 10.3390/cancers14061544

**Published:** 2022-03-17

**Authors:** Sabrina Jotschke, Susann Schulze, Nadja Jaekel, Beatrice Ludwig-Kraus, Robby Engelmann, Frank Bernhard Kraus, Christina Zahn, Nicole Nedlitz, Gabriele Prange-Krex, Johannes Mohm, Bettina Peuser, Maik Schwarz, Claudia Spohn, Timo Behlendorf, Mascha Binder, Christian Junghanss, Sebastian Böttcher, Haifa Kathrin Al-Ali

**Affiliations:** 1Krukenberg Cancer Center Halle, University Hospital Halle (Saale), 06120 Halle (Saale), Germany; sabrina.jotschke@uk-halle.de (S.J.); susann.schulze2@uk-halle.de (S.S.); nicole.nedlitz@uk-halle.de (N.N.); 2University Clinic and Outpatient Clinic for Internal Medicine IV, University Hospital Halle (Saale), 06120 Halle (Saale), Germany; nadja.jaekel@uk-halle.de (N.J.); christina.zahn@uk-halle.de (C.Z.); mascha.binder@uk-halle.de (M.B.); 3Central Laboratory, University Hospital Halle (Saale), 06120 Halle (Saale), Germany; beatrice.ludwig-kraus@uk-halle.de (B.L.-K.); bernhard.kraus@uk-halle.de (F.B.K.); 4Clinic III—Hematology, Oncology, and Palliative Care, Rostock University Medical Center, 18057 Rostock, Germany; robby.engelmann@med.uni-rostock.de (R.E.); christian.junghanss@med.uni-rostock.de (C.J.); sebastian.boettcher@med.uni-rostock.de (S.B.); 5Gemeinschaftspraxis Mohm/Prange-Krex, 01307 Dresden, Germany; prange@onkopraxis-dresden.de (G.P.-K.); mohm@onkopraxis-dresden.de (J.M.); 6Internistisch-Onkologische Ärztegemeinschaft, 04179 Leipzig, Germany; praxis.peuser@t-online.de; 7Paracelsus Medizinisches Versorgungszentrum, Schwerpunktpraxis für Hämatologie und Onkologie, 08261 Schoeneck, Germany; maik.schwarz@pkd.de; 8Hämatologisch-Onkologische Gemeinschaftspraxis, 06110 Halle (Saale), Germany; praxis-spohn@web.de; 9Gemeinschaftspraxis für Hämatologie, Onkologie und Gastroenterologie, 06110 Halle (Saale), Germany; tbehlendorf@onkologie-halle.de

**Keywords:** SARS-CoV-2 vaccination, myeloid neoplasms, lymphoid neoplasms, seroconversion, anti-spike-IgG, T-cells, CD4^+^-cells, CD8^+^-cells

## Abstract

**Simple Summary:**

The kinetics of SARS-CoV-2 spike-protein antibodies and the cellular immune landscape following vaccination in patients with hematologic neoplasms are poorly understood. The aim of our prospective and longitudinal study, which included 398 adults, was to compare day 35 and day 120 anti-spike-IgG antibody and day 120 SARS-CoV-2-specific T-cell responses in patients with hematologic malignancies to a reference cohort. Although day 35 seroconversion in controls (98%) was higher compared to patients with myeloid (82%) and lymphoid (48%) neoplasms, substantial increases in day 120 seroconversion were seen in both the myeloid (97%) and lymphoid (66%) cohorts. Remarkably, spike-specific CD4^+^- and CD8^+^-cells in the lymphoid (71%/31%) and control (74%/42%) cohorts were comparable. We provide strong evidence of vaccine-elicited immunogenicity in most patients with hematologic malignancies. Both kinetics of seroconversion and cellular responses are crucial to determine which patients with hematologic malignancies will generate immunity. The findings have implications on public health policy regarding recommendations for SARS-CoV-2 booster doses.

**Abstract:**

Purpose: To assess humoral responses longitudinally and cellular immunogenicity following SARS-CoV-2-vaccination in patients with hematologic and oncologic malignancies receiving checkpoint-inhibitors. Methods: This prospective multicenter trial of the East-German-Study-Group-for-Hematology-and-Oncology, enrolled 398 adults in a two (patients; *n* = 262) to one (controls; *n* = 136) ratio. Pre-vaccination, day 35 (d35), and day 120 (d120) blood samples were analyzed for anti-spike antibodies and d120 IL-2^+^IFNγ^+^TNFα^+^-CD4^+^- and CD8^+^-cells. Laboratories were blinded for patients and controls. Results: Patients belonged to the myeloid (*n* = 131), lymphoid (*n* = 104), and checkpoint-inhibitor (*n* = 17) cohorts. While d35 seroconversion was higher in controls (98%) compared to patients (68%) (*p* < 0.001), d120 seroconversion improved across all patient cohorts [checkpoint-inhibitors (81% to 100%), myeloid (82% to 97%), lymphoid (48% to 66%)]. CD4^+^- and CovCD8^+^-cells in the lymphoid (71%/31%) and control (74%/42%) cohorts were comparable but fewer in the myeloid cohort (53%, *p* = 0.003 /24%, *p* = 0.03). In patients with hematologic malignancies, no correlation between d120 humoral and cellular responses was found. A sizeable fraction of lymphoid patients demonstrated T-cell responses without detectable spike-specific-IgGs. Conclusions: Evidence of vaccine-elicited humoral and/or cellular immunogenicity in most patients is provided. Both humoral and cellular responses are crucial to determine which patients will generate/maintain immunity. The findings have implications on public health policy regarding recommendations for SARS-CoV-2 booster doses.

## 1. Introduction

Published data indicate that mortality from SARS-CoV-2 infections in patients with cancer is mainly associated with general risk factors such as older age and comorbidities [1,2]. Patients with hematologic malignancies have a particular high risk of COVID-19 related death with ~40% mortality rate [3,4]. COVID-19 vaccines have been developed and deployed with remarkable speed [5,6,7,8]. However, patients with malignancy were excluded from pivotal vaccination trials. Recent data indicate that active cancer therapy receiving patients with solid tumors develop adequate antibody responses to vaccination (although the magnitude of these responses is diminished relative to control cohorts) [9,10]. Depending on the type and activity of the disease, there is accumulating evidence that humoral immunity up to 42 days after the second dose of SARS-CoV-2 vaccines is impaired in patients with hematologic malignancies, especially if they were treated with B-cell depleting therapies such as anti-CD20 antibodies [11,12,13,14,15,16,17,18,19,20,21,22,23,24,25,26]. These data raised concerns about the efficacy of vaccines in generating humoral immunity in patients with hematologic malignancies, particularly in those with lymphoid neoplasms. However, little is known to the durability of vaccine-elicited antibody responses in patients with hematologic neoplasms. Although neutralizing antibodies are important in vaccine-induced protection as evidenced by the correlations of antibody responses and clinical outcome of COVID-19 infections [27,28,29], growing evidence points towards an equally important role for T-cells [30,31,32,33]. Circulating SARS-CoV-2-specific CD8^+^ and CD4^+^ T-cells were identified in ~70% and 100% of COVID-19 convalescent patients, respectively [34]. These T-cell responses were shown to be associated with improved survival after infections even in patients with hematologic neoplasms [35,36]. Similarly, early polyfunctional spike protein-specific T-cell responses were described after COVID-19 vaccination [37,38,39,40]. Indeed, stable and functional CD8^+^-T-cell responses could be mobilized one week after prime vaccination with BNT162b2 when circulating CD4^+^-T-cells and neutralizing antibodies were still weakly detectable [40]. mRNA-based vaccination generated robust CD4^+^- and CD8^+^-T-cell responses could be generated, despite poor antibody responses, in patients with multiple sclerosis on anti-CD20 antibody therapy after mRNA-based vaccination [41]. Liebers, et al. detected spike protein-specific T-cells 17 days after the second dose of vaccine using an IFNγ ELISPOT in 29/50 (58%) lymphoma patients who had received anti-CD20 treatments [24]. Using an intracellular cytokine assay for IFNγ, TNFα and IL2, Harrington, et al. reported polyfunctional CD8^+^- and CD4^+^-T-cells of 35% and 75%, respectively, in 21 patients with *BCR-ABL1*-negative myeloproliferative neoplasms (MPN) after a median of 21 days after a single dose of BNT162b2 vaccine [15].

Despite the relatively small sample sizes, the lack of a predefined sample collection, and the lack of large reference cohorts, these initial results suggest that patients with hematologic malignancies with insufficient humoral responses might still benefit from vaccinations through cellular responses, considering that effective T-cell responses are essential for SARS-CoV-2 clearance [42]. Indeed, a significant portion of antigen-specific T-cell responses may be missed if only IFNγ–related readouts are used and IL-2 is not taken into consideration [43]. Further, little is known about the longevity of vaccine-elicited cellular immunogenicity in large cohorts of patients with hematologic neoplasms.

In order to investigate the kinetics of IgG responses and relationship to specific T-cell responses, days 35 and 120 vaccine-induced humoral and day 120 cellular responses in patients with hematologic and oncologic malignancies receiving checkpoint inhibitors (PD-L1-inhibitors) were compared to a reference cohort.

## 2. Materials and Methods

### 2.1. Study Design

ImV-HOng (OSHO#98) is a longitudinal, prospective, multicenter, non-interventional study which compared day 35 (d35) and day 120 (d120) vaccine-elicited spike protein-specific humoral and d120 T-cell responses between patients and controls. The trial was conducted from 17 March 2021 to 6 December 2021 across seven centers of the East-German-Study-Group-for-Hematology-and-Oncology (OSHO). The trial was approved by the Ethical Review Boards and registered at the Paul-Ehrlich Institute (NIS-584) and Deutsches Register Klinischer Studien (DRKS00027372). The study received a grant from the German Leukemia and Lymphoma Foundation.

### 2.2. Study Cohorts

The study population comprised adult individuals willing to receive a SARS-CoV-2 vaccination. Participants were enrolled per random sampling after written informed consent into two cohorts at a 1 (controls) to 2 (patients) ratio. The control group included individuals without active cancer in the last five years. In addition to patients with myeloid and lymphoid neoplasms, patients with solid tumors receiving PD-L1 inhibition were also eligible for enrollment in the patient cohort in order to assess the impact of PD-L1 inhibition on immune response after vaccination. The inclusion and exclusion criteria are shown in Table 1.

### 2.3. Outcomes

The primary outcome was d35 SARS-CoV-2 spike-specific antibody concentrations in patients compared to controls following the first vaccination dose. The WHO launched the first International Standard for anti-SARS-CoV-2 immunoglobulin (IgG), wherein the neat sample was assigned to contain 1000 binding antibody units (BAU)/mL [44,45], and BAU/mL were subsequently converted to U/mL (U/mL = 0.972 × BAU/mL). Key secondary outcomes were d120 spike-specific humoral and T-cell responses in patients compared to controls. Baseline patient-, disease-, vaccination-, and laboratory-characteristics and any potential associations with vaccine-elicited responses were explored.

### 2.4. Procedures

Blood samples were drawn up to three weeks prior to vaccination, on d35 (±7), and d120 (±14) after the first vaccination dose. The pseudonymized samples were serially analyzed for SARS-CoV-2 spike-specific-IgGs in the Central Laboratory of the University Hospital Halle (Saale). T-cell responses were analyzed at the Special Hematology Laboratory, Rostock University Medical Center. Laboratories were blinded for patient and control groups.

### 2.5. Laboratory Measurements

#### 2.5.1. Measurement of SARS-CoV-2 Spike Protein Antibodies

The quantitative determination of IgG antibodies to the SARS-CoV-2 spike protein was carried out using the Roche Elecsys^®^ Anti-SARS-CoV-2 S assay (Roche Diagnostics International Ltd., Rotkreuz, Switzerland). The assay is based on a recombinant protein representing the receptor binding domain of the spike antigen in a double-antigen sandwich assay format, with a high specificity and sensitivity [46]. Antibody titers were measured on a Roche Cobas e 801 analyzer integrated in a fully automated Roche Cobas 8000 platform. A concentration of IgG SARS-CoV-2 spike protein antibodies of >0.8 U/mL is considered positive.

#### 2.5.2. SARS-CoV-2 Spike-Specific T-Cell Response

Heparinized whole blood was either left unstimulated (negative control), stimulated with 0.5 µg/mL Staphylococcus enterotoxin B (SEB, positive control) or stimulated using 0.6 nmol of (approximately 1 µg) wild-type spike protein of SARS-CoV2 peptides (SARS-CoV2 Prot_S Complete, REF: 130-127-953, Miltenyi Biotec [MB], Bergisch Gladbach, Germany) per ml blood for 4 h at 37 °C in the presence of Breveldin A. After incubation, bulk lysis, surface and intracellular staining were performed according to EuroFlow guidelines [47].

The panel comprised the following antibodies IL-2:BV421 (clone: MQ1-17H12, Biolegend, San Diego, CA, USA), CD45RA:VioGreen (clone: REA1047), CCR7:FITC (clone: REA546), IFNγ:PE (clone: 45-15), CD4:PE-Vio615 (clone: REA623), CD8:PE-Vio770 (clone: REA734), TNFa-APC (clone: REA656), and CD3:APC-Vio770 (clone: REA613) that were purchased from MB, unless stated otherwise. A median 2,660,024 nucleated cells per sample were acquired on Becton Dickinsion (FACS Lyric) or MB (MACS Quant) flow cytometers. Primary data were analyzed in Infinicyt (v2.0.4b, Cytognos SL, Salamanca, Spain). Gating was in line with recommended standards for ICS assays [48,49].

Raw event numbers and frequencies per population were exported and analyzed using R (v4.1.1). Normalized percentages of SEB-activated and spike-specific T-cells were calculated by subtracting the respective frequencies of the negative control measured for the same sample and expressed as percentage of total CD4^+^ and CD8^+^ T-cells of the sample [37,38]. A cohort of 14 not vaccinated and self-reportedly non-infected controls was used to calculate the limit of detection as follows: the z-score for each control sample was calculated per parameter. Samples with a z-score above two were considered as outlier for that parameter and removed (one outlier per parameter was detected). The limit of detection (LOD) was calculated as mean +2SD. All samples above the LOD [0.00459% for CD4 + IL-2 + IFNγ + TNFα + (CovCD4) and 0.00287% for CD8 + IL-2 + IFNγ + TNFα + (CovCD8) T-cells] were considered positive.

### 2.6. Statistical Analysis

Sample size was calculated based on published data to the immune response after 30 μg BNT162b2 (Comirnaty ©Biontech/Pfizer) vaccine [37]. Assuming a standard deviation of 0.9 for the logarithm of geometric mean concentrations, enrollment of 236 and 118 evaluable patients and controls respectively would provide 80% power (alpha error, 5%) to detect a significant difference in d35 seroconversions between patients and controls.

Continuous covariates were summarized as medians and interquartile ranges (IQRs) and categorical parameters as absolute and relative frequencies. Humoral responses (i.e., anti-Spike IgG concentrations > 0.8 U/mL) on d35 and d120 were compared between patients and controls by evaluating the mean difference in concentrations using t-tests and reporting the 95% confidence interval (CI). Cellular responses on d120 (i.e., CovCD4^+^ and CovCD+ above the LOD) were similarly compared and expressed. Vaccine-elicited seroconversion and cellular response rates in patient cohorts (i.e., type of diagnosis; cancer therapy vs. none) were evaluated in subgroup analyses. Regression models were used to test the association of baseline characteristics with vaccine-induced humoral and cellular responses. Baseline patient-related factors included age [continuous variable, 5- and 10-years frequency-matching], gender, baseline B- and T-cell counts, pre-vaccination anti-spike-IgGs, and cohort category. Vaccine-related variables were type of vaccine, number of injections, and interval between injections (continuous variable; interval ≤35 vs. >35 days).

Secondary endpoint analyses were explorative. Statistical tests were two-tailed and *p* values < 0.05 were considered significant. Analyses were performed using IBM Corp. Released 2021. IBM SPSS Statistics for Windows, Version 28.0. Armonk, NY, USA: IBM Corp.

## 3. Results

### 3.1. Patient Characteristics

A total of 398 adults were enrolled [controls, *n* = 136; patients, *n* = 262]. Patients had myeloid (*n* = 135) and lymphoid (*n* = 108) neoplasms, and cancer under checkpoint inhibition (*n* = 19). A CONSORT-Flowchart of participants is shown in Figure 1. This analysis comprises 385 participants who actually received the first vaccination [patients *n* = 252 (96.2%); controls *n* = 133 (97.8%)]. Table 2 illustrates the characteristics of vaccinated participants. Patients in the myeloid cohort were most frequently diagnosed with *BCR-ABL1*-positive (*n* = 29) and negative myeloproliferative neoplasms (*n* = 57). Compared to controls, patients were older (*p* < 0.001). Prior to vaccination, 186 (76.2%) patients were on active cancer therapy. An allogeneic hematopoietic-cell-transplantation (HCT) was documented in 32 participants. The majority of participants (82.6%) received mRNA-based vaccines. A second dose was given to 230 (91.3%) patients and 107 (80.5%) controls after a median of 40 days for patients and 33 days for controls (*p* = 0.2). Reasons for only one injection were vaccination with the vector-based COVID-19 Vaccine Janssen by ©Johnson&Johnson (*n* = 21), a history of a SARS-CoV2 infection prior to vaccination (*n* = 17), and others (*n* = 10). Due to health authority guidelines, ~50% of participants received the second dose 42 days after the first. A history of a SARS-CoV2 infection prior to vaccination with a median of 7 months and a median pre-vaccination anti-spike-IgG concentration of 122 U/mL (IQR 23.9-480) was documented in 20 (5.2%) subjects. No antibodies were detected in one patient and one control. Anti-spike-IgGs prior to vaccination were detected in 11 participants (9 patients and 2 controls) with no history of a previous infection.

### 3.2. SARS-CoV-2 Spike-Specific Humoral Response

#### 3.2.1. Day 35 Spike-Specific Seroconversion

Though no difference in d35 anti-spike-IgG mean values between patients and controls (95% CI, −1438.5–559.8) were detected, anti-spike-IgGs >0.8 U/mL were measured in 121 (98%) controls and 162 (68%) patients (*p* < 0.001). Seroconversion rates in both the myeloid (82%) and lymphoid cohorts (48%) were lower compared to controls (*p* < 0.001) (Table 3; Figure 2). The same trends were seen when median titers were considered. Seroconversion occurred in 13 (81%) patients under checkpoint inhibitor therapy.

#### 3.2.2. Day 120 Spike-Specific Seroconversion

The difference in d120 mean IgG values between patients and controls was not significant (95% CI, −639.4–889.1). However, mean values on d120 were significantly higher across all study participants compared to d35 with a mean difference of 477 U/mL (95% CI, 92.4–861.6). Seroconversion in controls was maintained (98%) and substantial increases in the myeloid (97%) and checkpoint inhibitor (100%) cohorts were seen. These response rates were higher compared to those seen in the lymphoid group (66%) (*p* < 0.001). Similarly, median IgG levels were highest in controls (1212 U/mL) and lowest in the lymphoid cohort (88.3 U/mL) (*p* < 0.001) (Table 3). Overall, 76% of controls and patients maintained humoral immunogenicity over time (Figure 2). An association between d35 and d120 anti-spike-IgGs in 333 paired samples [controls (*n* = 117); patients (*n* = 216)] was found (*R*^2^ = 0.34; *p* < 0.001). Seroconversion on d120, despite d35 IgGs < 0.8 U/mL was documented in 18/20 (90%) and 22/49 (45%) patients with myeloid and lymphoid neoplasms respectively. Five of six subjects who lost d35 response belonged to the lymphoid group. Anti-spike-IgGs > 0.8 U/mL on d120 were detected in 30/33 (91%) subjects including 14 patients who received only one vaccine injection.

### 3.3. Day 120 SARS-CoV-2 Spike-Specific T-Cell Response

CovCD4 and/or CovCD8 were detected in 155/223 (69.5%) subjects [controls: 81.5%; patients 61% (*p* = 0.02)] with CovCD4 being more frequent than CovCD8 responses (*p* < 0.001) (Table 3, Figure 3). The differences in mean values of CovCD4 (95% CI, −0.00–0.02) and CovCD8 (95% CI, −0.00–0.02) between patients and controls were not significant. CovCD4 cells were more frequently detected in the control (74%) and lymphoid (71%) cohorts compared to the myeloid cohort (53%, *p* = 0.003 vs. controls), mirrored by a similar trend for CovCD8 cells (myeloid 28%, controls 42%, *p* = 0.03, Table 3). For controls, significant, but relatively weak pair-wise correlations were seen between d120 IgG responses, CovCD4, and CovCD8 cells (Figure 3). Such a correlation could not be detected in the myeloid or lymphoid cohorts. A sizeable fraction of patients in the lymphoid cohort demonstrated CovCD4 and/or CovCD8 responses without detectable spike-specific IgGs (Figure 3).

### 3.4. Predictors of Spike-Specific Immune Responses

Both patient- and vaccine-related factors were evaluated for a potential impact on humoral and cellular immune responses. Patient-related variables included age [continuous variable, 5- and 10-years frequency-matching], gender, baseline B- and T-cell counts, pre-vaccination anti-spike-IgGs, cohort category [controls vs. patients], and diagnosis [myeloid neoplasm vs. lymphoid neoplasm vs. solid tumor receiving PD-L1 inhibition]. Vaccine-related factors were the type of vaccine, number of injections (one versus two), and interval between injections [continuous variable; interval ≤35 vs. >35 days].

For the entire study cohort, with the exception of pre-vaccination anti-spike-IgGs (*R*^2^ = 0.2; *p* < 0.001), no relationship was found between d35 and d120 humoral responses and patient-related variables, including older age and vaccine-related factors. In the myeloid group, only pre-vaccination anti-spike-IgGs were associated with higher d35 and d120 humoral responses. The interval between injections had no significant impact on vaccine-induced humoral or cellular responses. Although statistically not significant (*p* = 0.05), an interval >35 days between injections tended to be associated with lower d35 but not d120 humoral responses in the lymphoid cohort only. In the myeloid and lymphoid cohorts, CovCD4 and CovCD8-cell responses were not associated with patient- or vaccine-related variables.

Due to the diversity of cancer therapies, the impact of treatment on vaccine-elicited responses was explored without regression models. For the myeloid cohort, d35 (83%) and d120 (97.4%) humoral as well as CovCD4 (53%) and CovCD8 (28%) responses were comparable in patients receiving tyrosine-kinase-inhibitors for *BCR-ABL1*-positive CML, JAK-inhibitors for myeloproliferative neoplasms, and other therapies to those on no treatment (*p* = 0.3). For the lymphoid group, d35 seroconversion was lowest in the B-cell depleting therapy (13%) or bruton-tyrosine-kinase-inhibitor (BTKi) (21.4%) groups compared to other (58%) or no (62.5%) treatment groups (*p* < 0.001). No patient on B-cell depleting therapy had detectable d120 anti-spike-IgGs compared to positive seroconversions in the BTKi [7/15; (47%)], other treatment [23/30; (77%)], and no therapy [33/38; (87%)] groups (*p* < 0.001). CovCD4 [30/42 (71.4%)] and CovCD8 [13/42 (31%)] were detected across all treatment categories.

Humoral responses on d35 were documented in 21/30 (70%) subjects with a history of HCT. On d120, 8/9 subjects with negative d35 antibodies seroconverted including 3/4 patients with a HCT-to-vaccination interval <12 months. CovCD4 (62.5%) and CovCD8 (29%) were measured after HCT.

## 4. Discussion

Despite the older age of patients compared to controls, sustainable and/or improvements in seroconversion rates and anti-splike-IgG concentrations over time were observed across all cohorts. As expected, d35 seroconversion was higher in controls (98%) compared to patients (68%) (*p* < 0.001). However, d120 seroconversion improved across all patient cohorts [oncologic malignancies under PD-L1 inhibitor therapies (81% to 100%), myeloid neoplasms (82% to 97%), lymphoid neoplasms (48% to 66%)]. Indeed, patients with myeloid and oncologic malignancies under PD-L1 inhibitor therapies had comparable seroconversion rates to the control group. The few cases with pre-vaccination anti-spike-IgGs without a known history of COVID-19 infection might represent asymptomatic infections or cross-reactive antibodies generated during previous infections with other coronaviral strains [50].

Another key finding was the remarkable and largely seroconversion-independent d120 SARS-CoV-2–specific CD4^+^TNFa^+^IFNγ^+^IL-2^+^- and CD8^+^TNFa^+^IFNγ^+^IL-2^+^-cells across all cohorts, particularly in the lymphoid group with the lowest seroconversion rate. Indeed, the cellular response in patients with lymphoid neoplasms with detectable CovCD4^+^- in 71% and CovCD8^+^-cells in 31% of cases was comparable to that measured in the control group (CovCD4^+^- 74% and CovCD8^+^-cells 42%). A sizeable fraction of lymphoid patients demonstrated T-cell responses without detectable spike-specific-IgGs. Overall, CD4^+^ T-cell responses outnumbered CD8^+^ responses in our study. This is in line with what has been observed in immunocompetent individuals [34,51].

To our knowledge, this work is the first to describe the kinetics of SARS-CoV-2 vaccine-induced humoral and cellular responses over time. The impaired early (d35) seroconversion in patients with hematologic malignancies is in line with previous publications [11,12,13,14,15,16,17,18,19,20,21,22,23,24,25,26]. However, the majority of patients demonstrate sustained and/or improved humoral and/or cellular responses if measured later (d120). These immune responses were seen irrespective of the type of vaccine or interval between injections. In a recent longitudinal study, antibodies against the SARS-CoV-2 spike antigen and specific memory cell responses were detected in 96% and 63% of health care workers four- and eight-months post infection [52,53].

In line with the literature, our data imply that the previously reported “early“ T-cell responses [15,24,34,35,36,37,38,39,40,41] are likely to persist for several months in patients with hematologic malignancies after vaccination similar to what has been observed in immunocompetent individuals after COVID-19 infections [30,31,32,53].

Taken together, our results underscore the need for large-scale follow-up data to establish standardized post-vaccination time-windows for humoral and cellular response assessments to identify “true vaccination failures” in cancer patients.

However, the routine applicability of tests to measure humoral and cellular immune responses remains challenging. Although several assays for anti-SARS-CoV-2-IgG are commercially available, current assays generate discrepant results. In fact, we are still far from the identification of optimal thresholds for IgG-positivity as a surrogate for neutralization capacity and neutralizing antibodies (NAbs) which confer protection [29,54,55,56,57]. Further, correlations between NAbs and clinical efficacy against infections are weak and likely rely on the population tested [58,59].

The issue is even more complicated regarding cellular response assays. Generally, they are not readily available and mainly used for research purpose. There is often a preponderance of using IFNγ–related readouts to assess T-cell responses [24,60]. However, data suggest that polyfunctional T-cells have higher protective efficacy after vaccination compared to IFNγ monofunctional T-cells [61]. The true percentages of patients developing polyfunctional vaccine-induced CD4^+^TNFa^+^IFNγ^+^IL-2^+^- and CD8^+^TNFa^+^IFNγ^+^IL-2^+^-cells might be incorrectly assessed if IL-2 is not considered [43].

Although cellular responses are promising indicators of immunity, our data do not suggest that those with a response compared to those without such a response are more likely to be protected. Yet, even if infections cannot be prevented, it is still possible that T-cell responses are sufficient to ensure mild courses of COVID-19 disease. Thus, studies are necessary to evaluate the degree of cellular-induced clinical protection. Further, SARS-CoV-2 variants such as Omicron (B.1.1.529) with their antibody escape highlight the importance of addressing whether T-cell recognition is also affected.

One limitation is that we did not measure NAbs with virus neutralization assays which are considered to be the gold standard. Yet, we used an anti-SARS-CoV-2 IgG assay with cutoffs for reasonable prediction of NAb [62]. After enrollment started, health authorities in some federal states in Germany changed the interval between injections from 21 to 42 days. The potential impact on d35 response evaluation was discussed with the statistician and accounted for by including the interval as a vaccine-related variable in the regression model. As ~50% of participants across all cohorts received the second injection 42 days later, comparison between groups was feasible. Finally, despite the relatively large number of participants, data of secondary outcomes remain explorative and need to be confirmed in larger trials.

In summary, our longitudinal study describes the nature of SARS-CoV-2 vaccine-induced humoral and cellular immune landscape in patients with hematologic on oncologic malignancies under PD-L1 inhibition.

## 5. Conclusions

We provide strong empirical evidence of early and late SARS-CoV-2 vaccine-elicited immunogenicity in patients with hematologic neoplasms and oncologic patients receiving checkpoint inhibitors. Even with blunted and heterogeneous antibody responses, T-cell priming seems to be largely intact. This study provides key information and fills knowledge gaps with respect to T-cell responses in vulnerable persons. Both the kinetics of anti-SARS-CoV-2 antibodies over time and cellular responses are crucial to determine which patients will generate and maintain immunity after vaccination.

The findings have implications on clinical decision-making for designing vaccine strategies given the current timing and recommendations for SARS-CoV-2 booster doses. It will be important in the future to determine whether the residual humoral immunity and sustained T-cell responses, retain the ability to respond to emerging SARS-CoV-2 variants. Larger studies with clinical outcomes are needed.

## Figures and Tables

**Figure 1 cancers-14-01544-f001:**
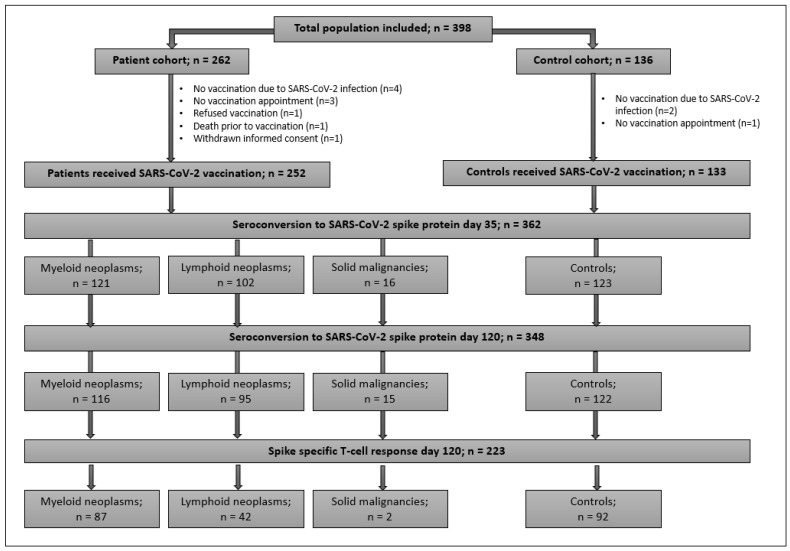
CONSORT flowchart of study population.

**Figure 2 cancers-14-01544-f002:**
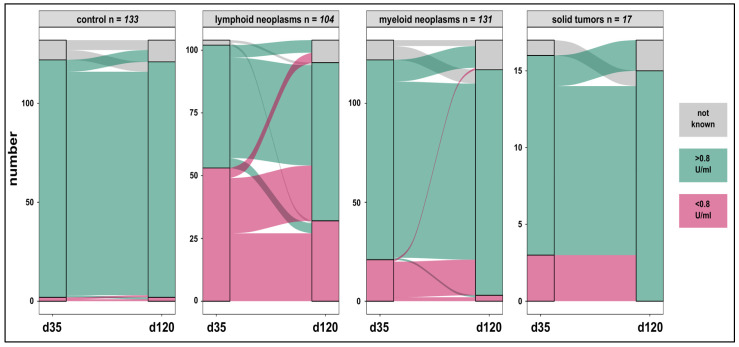
Humoral anti-spike-specific responses in patients and controls on day 35 and day 120 after vaccination. Specific IgG responses were maintained at high rates in controls (98%) and increased in patients with oncologic malignancies on checkpoint inhibitors (81% to 100%), lymphoid (48% to 66%), and myeloid neoplasms (82% to 97%). A complete loss of the anti-spike IgG was rarely seen.

**Figure 3 cancers-14-01544-f003:**
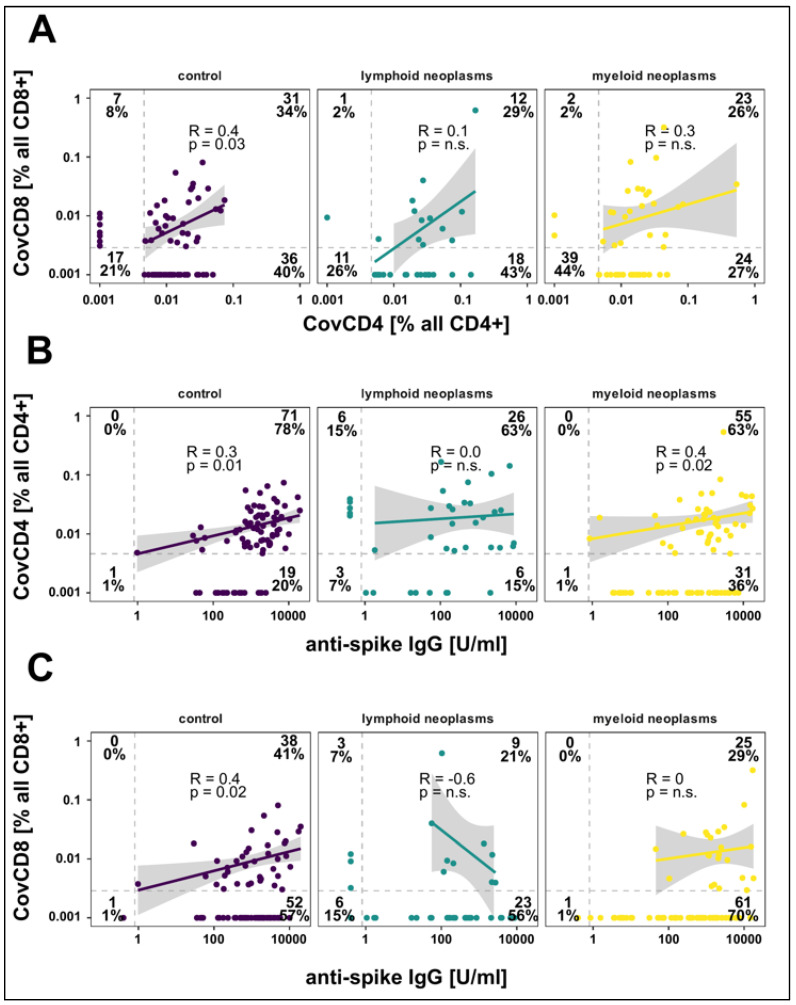
Correlation between spike-specific CD4 + IL-2 + IFNγ + TNFα + (CovCD4) and CD8 + IL-2 + IFNγ + TNFα + (CovCD8) cell responses (**A**) and anti-spike IgG concentrations (**B**,**C**) on day 120 after vaccination. Results are shown for controls as well as for patients with lymphoid and myeloid neoplasms. Broken lines represent the limit of detection (LOD). Regression lines, Spearman correlation coefficients, and significance are calculated for double positive patients.

**Table 1 cancers-14-01544-t001:** Inclusion and exclusion criteria.

	Patients	Controls
Inclusion criteria	Age ≥ 18 years Presence of one of the following diagnoses: - myeloid neoplasm - lymphoid neoplasm - solidtumor under PD-L1 * inhibition Willing to receive a SARS-CoV-2 vaccination	Age ≥ 18 years No active malignancy in the last 5 years Willing to receive a SARS-CoV-2 vaccination
Exclusion criteria	Contraindication to a SARS-CoV-2 vaccination Limited legal capacity to consent	Contraindication to a SARS-CoV-2 vaccination Limited legal capacity to consent

* PD-L1, programmed death ligand 1.

**Table 2 cancers-14-01544-t002:** Baseline characteristics of vaccinated study population.

Parameter		Total Patient Cohort	Myeloid Neoplasms	Lymphoid Neoplasms	Solid Tumors	Controls
		*n* = 252	*n* =131	*n* = 104	*n* = 17	*n* = 133
Age (y)	median (IQR)	62 (52–71)	61 (52–68)	66 (51–74)	61 (57–68)	54 (42–67)
≥60 years	*n* (%)	146 (57.9)	70 (53.4)	66 (63.5)	10 (58.8)	49 (36.8)
Gender, male	*n* (%)	139 (55.2)	69 (52.7)	60 (57.7)	10 (58.8)	54 (40.6)
**Diagnosis**						N/A
MPN	*n* (%)		91 (69.5)	N/A	N/A	
AML	*n* (%)		10 (7.5)	N/A	N/A	
MDS	*n* (%)		15 (11.5)	N/A	N/A	
Lymphoma	*n* (%)		N/A	40 (38.5)	N/A	
CLL	*n* (%)		N/A	32 (30.8)	N/A	
Multiple myeloma	*n* (%)		N/A	22 (21.2)	N/A	
Others	*n* (%)		15 (11.5)	10 (9.6)	17 (100)	
**Baseline lab values**						N/A
WBC < LLN of 3.7 × 10^9^/L	*n* (%)	25/193 (13)	12/105 (11.4)	12/74 (16.2)	1/14 (7.1)	
Granulocytes < LLN of 1.8 × 10^9^/L	*n* (%)	19/177 (10.7)	7/101 (6.9)	12/66 (18.2)	0	
Lymphocytes < LLN of 1.1 × 10^9^/L	*n* (%)	56/180 (31.1)	30/102 (29.4)	21/68 (30.9)	5/10 (50)	
B-cells < LLN of 73/µL	*n* (%)	26/109 (23.9)	12/66 (18.2)	12/39 (30.8)	2/4 (50)	
T-cells < LLN of 856/µL	*n* (%)	41/109 (37.6)	23/66 (34.8)	15/39 (38.5)	3/4 (75)	
CD4^+^ T-cells < LLN of 491/µL	*n* (%)	49/108 (45.4)	25/66 (37.9)	21/38 (55.3)	3/4 (75)	
CD8^+^ T-cells < LLN of 162/µL	*n* (%)	21/108 (19.4)	15/66 (22.7)	4/38 (10.5)	2/4 (50)	
LDH > ULN of 4.2 µkat/L	*n* (%)	53/175 (30.3)	35/98 (35.7)	12/64 (18.8)	6/13 (46.2)	
**Prior COVID-19 infection**	*n* (%)	11 (4.4)	5 (3.8)	6 (5.8)	0 (0)	9 (6.8)
**Type of vaccine ***						
mRNA-based	*n* (%)	215 (85.3)	108 (82.5)	91 (87.5)	16 (94.1)	103 (77.4)
Vector-based	*n* (%)	35 (13.9)	21 (16)	13 (12.5)	1 (5.9)	29 (21.8)
Missing	*n* (%)	2 (0.8)	2 (1.5)	0 (0)		1 (0.8)
**Interval between 1st and 2nd vaccination**		*n* = 230	*n* = 118	*n* = 95	*n* = 17	*n* = 107
Median Interval	d (IQR)	40 (22–42)	40 (22–42)	42 (21–42)	29 (21–42)	33 (21–42)
≤35 days	*n* (%)	110 (47.8)	55 (46.6)	45 (47.4)	10 (58.8)	58 (54.2)
>35 days	*n* (%)	120 (52.2)	63 (53.4)	50 (52.6)	7 (41.2)	49 (45.8)
**Active oncologic therapy**	*n* (%)	171 (67.9)	91 (69.5)	63 (60.6)	17 (100)	N/A
**Type of active oncologic therapy**						
TKI	*n* (%)		44 (48.3)	N/A	N/A	
INF	*n* (%)		9 (9.9)	N/A	N/A	
BTK-inhibitor	*n* (%)		N/A	15 (23.8)	N/A	
B-cell-depleting therapy	*n* (%)		N/A	15 (23.8)	N/A	
Chemotherapy	*n* (%)		18 (19.8)	6 (9.5)	N/A	
Checkpoint inhibitor	*n* (%)		N/A	1 (1.6)	17 (100)	
Others	*n* (%)		20 (22)	26 (41.3)	N/A	

* mRNA-based vaccines: BNT162b2 Comirnaty ©Biontech/Pfizer or mRNA-1273 vaccine ©Moderna; vector-based vaccines: Vaxzevria ©AstraZeneca or COVID-19 Vaccine Janssen by ©Johnson&Johnson. Abbreviations: AML, acute myeloid leukemia; BTK, bruton tyrosine kinase; CLL, chronic lymphatic leukemia; d, days; INF, interferone; IQR, interquartile range; LDH, lactatdehydrogenase; LLN, lower limit of normal; MDS, myelodysplastic syndrome; MPN, myeloproliferative neoplasm; N/A, not applicable; TKI, tyrosine kinase inhibitor; ULN, upper limit of normal; WBC, white blood cell count.

**Table 3 cancers-14-01544-t003:** Humoral and T-cell response to vaccination in controls and patient cohorts.

Parameter		Controls	Myeloid Neoplasms	Lymphoid Neoplasms	Solid Tumors
Anti-spike IgG					
day 35	n (%)	121 (98%)	100 * (82%)	49 *^,#^ (48%)	13 (81%)
	Median (IQR) [U/mL]	166 (32–1558)	27.9 ^§^ (2.4–466)	0.59 ^§,&^ (0.39–30)	22.4 ^§^ (1.1–462)
day 120	n (%)	120 (98%)	113 (97%)	63 *^,#^ (66%)	15 (100%)
	Median (IQR) [U/mL]	1212 (506–2854)	874 ^§^ (149–2063)	88.3 ^§,&^ (0.39–535)	130 ^§^ (52–1153)
**SARS-CoV2 specific T cells**					
CovCD4^+^	n (%)	68 (74%)	46 * (53%)	30 ^#^ (71%)	−
	Median (IQR) [% of CD4^+^ T cells]	0.0091 (0.0044–0.0189)	0.0057 (0.0015–0.0178)	0.012 ^&^ (0.0039–0.029)	−
CovCD8^+^	n (%)	39 (42%)	24 * (28%)	13 (31%)	−
	Median (IQR) [% of CD8^+^ T cells]	0.0016 (0.0005–0.0064)	0.0031 ^§^ (0.0001–0.0035)	0.0011 (0.0001–0.0039)	−

Numbers (and %) of subjects with IgG concentrations >0.8 U/mL, CovCD4^+^, and CovCD8^+^ responses above the limit of detection (LOD) per parameter. Medians and interquartile ranges (IQR) by parameter and cohort. *^,§^: significantly different from control; ^#,&^: significantly different from myeloid neoplasms.

## Data Availability

The trial was registered at Deutsches Register Klinischer Studien (DRKS00027372) and the Paul-Ehrlich Institute (NIS-584).

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
