# Peer review of "Longitudinal Humoral and Cellular Immune Responses Following SARS-CoV-2 Vaccination in Patients with Myeloid and Lymphoid Neoplasms Compared to a Reference Cohort: Results of a Prospective Trial of the East German Study Group for Hematology and Oncology (OSHO)"

_cancers, 2022, doi:10.3390/cancers14061544_

Round 1
Reviewer 1 Report
The authors present novel data on humoral and T cell responses to SARS-COV-2 vaccination in patients with haematological and oncological malignancies with consideration for responses in patients receiving check point inhibitors.
Overall the data is valuable and adds to the body of knowledge in this field. The manuscript would be strengthened by the following considerations:
- The methods are sound but require more clarity. For example why only patients with oncological malignancies with PDL1 inhibition were selected. This is certainly an area with a data gap but the rationale for this could be clearer. T
- he methods section would read easier if inclusion exclusion criteria are clearly listed.
- It is not clear how the study accounted for patients who may have had prior infections with SARS-COV-2 that could have confounded the humoral and T-cell response assessments used in the study.
- The justification for the use of the testing strategies for humoral and cellular responses is sound. A discussion of the applicability of these tests is needed before the conclusions around the implications of this study on public health.
- The paragraph starting line 352 over states the size of the sample as the diseases subgroups sample samples are small and likely too small to make conclusions about the efficacy of vaccination in the diseases included in the study.
- The manuscript would benefit from editing to make it easier to read and to correct grammatical and syntax errors. The authors could consider refraining from using the term “hematologic or oncologic patients”. The preferred term by patients would be patients with hematologic or oncologic malignancies.
Author Response
Dear Reviewer,
thank you very much for reviewing our manuscript. We would like to resubmit a revised version In this cover letter we would list all amendments and responses to the points raised by the reviewers and how we have dealt with them in the manuscript.
Reviewer 1
The authors present novel data on humoral and T cell responses to SARS-COV-2 vaccination in patients with haematological and oncological malignancies with consideration for responses in patients receiving check point inhibitors.
Overall the data is valuable and adds to the body of knowledge in this field. The manuscript would be strengthened by the following considerations:
Point 1: The methods are sound but require more clarity. For example why only patients with oncological malignancies with PDL1 inhibition were selected. This is certainly an area with a data gap but the rationale for this could be clearer.
Response: in addition to patients with hematologic malignancies, the protocol was designed to allow the inclusion of patients with solid tumors receiving PDL1 inhibition to explicitly assess the impact of checkpoint inhibition on immune response after vaccination as indeed there is a data gap to this issue. Other treatments for solid tumors were excluded in the protocol to avoid heterogeneity.
To clarify the rationale behind this, the following sentence was included under point 2.2 (Study cohorts):
In addition to patients with myeloid and lymphoid neoplasms, patients with solid tumors receiving PD-L1 inhibition were also eligible for enrollment in the patient cohort in order to assess the impact of PD-L1 inhibition on immune response after vaccination.
Point 2: The methods section would read easier if inclusion exclusion criteria are clearly listed.
Response: Under point 2.2 (Study cohorts), the following sentence and table were included:
The inclusion and exclusion criteria are shown in Table-1.
Table 1. Inclusion and exclusion criteria
|
Patients |
Controls |
Inclusion criteria |
l Age ≥ 18 years l Presence of one of the following diagnoses: - myeloid neoplasm - lymphoid neoplasm -solid tumor under PD-L1* inhibition l Willing to receive a SARS-CoV-2 vaccination |
l Age ≥ 18 years l No active malignancy in the last 5 years
l Willing to receive a SARS-CoV-2 vaccination
|
Exclusion criteria |
l Contraindication to a SARS-CoV-2 vaccination l Limited legal capacity to consent |
l Contraindication to a SARS-CoV-2 vaccination l Limited legal capacity to consent |
*PD-L1=Programmed death ligand 1
Point 3: It is not clear how the study accounted for patients who may have had prior infections with SARS-COV-2 that could have confounded the humoral and T-cell response assessments used in the study.
Response: Previous publications excluded patients with prior infections. Yet, the aim of our trial was to reflect a real world scenario. Thus, a previous SARS-COV-2 infection was not an exclusion criterion. This was already accounted vor while designing the protocol by including a pre-vaccination anti-spike-IgG measurement as the number of patients with a history or even asymptomatic infections that will actually be enrolled was not known at that time.
Per protocol, these pre-vaccination values were included as a statistical variable for the humoral and T-cell response assessments as described in point 2.6. „Statistical Analysis“ and their impact on humoral and T-cell responses are presented under point 3.4. „Predictors of spike-specific immune responses“.
Overall and as shown in table 2, only 11 patients (myeloid=5; lymphoid=6; PD-L1=0) and 9 controls had a history of a previous Covid-19 infection.
For clarity, the following sentence has now been included at the beginning of point 3.4. „Predictors of spike-specific immune responses“ to give readers a complete overview of all parameters tested:
Both patient- and vaccine-related factors were evaluated for a potential impact on humoral and cellular immune responses. Patient-related variables included age [continuous variable, 5- and 10-years frequency-matching], gender, baseline B- and T-cell counts, pre-vaccination anti-spike-IgGs, cohort category [controls vs patients], and diagnosis [myeloid neoplasm vs lymphoid neoplasm vs solid tumor receiving PD-L1 inhibition]. Vaccine-related factors were the type of vaccine, number of injections (one versus two), and interval between injections [continuous variable; interval ≤ 35 vs >35 days].“
Point 4: The justification for the use of the testing strategies for humoral and cellular responses is sound. A discussion of the applicability of these tests is needed before the conclusions around the implications of this study on public health.
Response: The discussion has been re-edited as suggested. the wording „and public health policy“ in Conclusions has been deleted and the following sentence included:
„Larger studies with clinical outcomes are needed.“
In the Discussion, the paragraph to the applicability reads now as follows:
„Taken together, our results underscore the need for large-scale follow-up data to establish standardized post-vaccination time-windows for humoral and cellular response assessments to identify „true vaccination failures“ in cancer patients. However, the routine applicability of tests to measure humoral and cellular immune responses remains challenging. Although several assays for anti-SARS-CoV-2-IgG are commercially available, current assays generate discrepant results. In fact, we are still far from the identification of optimal thresholds for IgG-positivity as a surrogate for neutralization capacity and neutralizing antibodies (NAbs) which confer protection [29,57-60]. Further, correlations between NAbs and clinical efficacy against infections are weak and likely rely on the population tested [61,62].
The issue is even more complicated regarding cellular response assays. Generally, they are not readily available and mainly used for research purpose. There is often a preponderance of using IFNγ–related readouts to assess T-cell responses [24,63]. However, data suggest that polyfunctional T-cells have higher protective efficacy after vaccination compared to IFNγ monofunctional T-cells [64]. The true percentages of patients developing polyfunctional vaccine-induced CD4+TNFa+IFNγ+IL-2+- and CD8+TNFa+IFNγ+IL-2+-cells might be incorrectly assessed if IL-2 is not considered [43].“
Point 5: The paragraph starting line 352 over states the size of the sample as the diseases subgroups sample samples are small and likely too small to make conclusions about the efficacy of vaccination in the diseases included in the study.
Response: the following sentence has been deleted:
The large sample size, statistical power, multicenter nature, and the representative older age of patients allow meaningful inferences.
Point 6: The manuscript would benefit from editing to make it easier to read and to correct grammatical and syntax errors. The authors could consider refraining from using the term “hematologic or oncologic patients”. The preferred term by patients would be patients with hematologic or oncologic malignancies.
Response: thanks to the valuable comments of the reviewers, the manuscript is now hopefully easier to read. The Discussion section is completely re-edited for an easier reading. It now starts with a paragraph summarizing the results and their novelty, a paragraph on the potential implication of the findings, a paragraph to challenges around the applicability of tests for immunity, a paragraph relating to other studies on immunogenicity of COVID vaccines in cancer patients and how our results compare to these studies, and a paragraph on the limitations of the study.
The manuscript has been checked by a native English-speaking colleague and the phrasing "hematologic or oncologic patients" has been replaced by "patients with hematologic or oncologic malignancies"according to the proposal.
We hope that we have adequately answered the points raised by the reviewer
Sincerely
Yours
Susann Schulze and coauthors
Reviewer 2 Report
This is an interesting article dealing with a COVID19 vaccine elicited immunological responses in patients with hematological malignancies. It goes beyond measuring antibody responses and looks at T cell responses.
Some suggestions to the authors:
Line 72: This statement is not accurate. The references cited report cohorts of cancer patients with symptomatic COVID 19, and found that risk factors for COVID19 mortality were similar to those of the general population. Except for patients with hematological malignancies, there is no clear evidence that cancer, in itself, is a risk factor for sever COVID19.
Line 126: Were cases and controls matched for age and gender ?
Line 186: what do you mean by “sample size was calculated based on the literature”? sample size should be based on power calculations, which should be based on the estimated difference between patients and controls, based on the literature. Please re-phrase.
Line 218: What were the reasons for not administering the 2nd dose of the vaccine? Did you include the participants that received only the 1st dose in the same analysis as those that received 2 doses ?
Line 257: The sentence “with a mean difference … participants” is unclear.
Figure 2: Please add the number of patients represented in each of the graphs.
Line 264: It is plausible that among patients with low IgG titers on d35 there would be a hogh percentage of participants that got only the 1st vaccine dose. Please clarify if there was an association between low IgG titers on D35 and receipt of the 2nd vaccine dose more than 35 days after the 1st dose (after testing). Some discussion on this question exists in the “discussion” section (line 362). I would suggest you clarify this discussion and move it to the “results” section.
Discussion – the discussion needs editing, both content-wise and language editing. It seems as if it was written by a different author then the rest of the manuscript. I suggest you re- write it and then have a linguistic editor make final corrections. The discussion section should include a paragraph summarizing the results and their novelty, a paragraph on the potential implication of your findings, a paragraph relating to other studies on immunogenicity of COVID vaccines in cancer patients and how your results compare to these studies, and a paragraph on the limitations of your study.
Conclusions: “The findings have implications on clinical decision-making and public health policy for designing vaccine strategies given the current timing and recommendations for SARS- 379 CoV-2 booster doses.” – this seems like an over- statement, since this study did not look at clinical results (infections/ complications of COVID19) and it’s cohorts comprise of 50-100 patients. I would say that the findings suggest that the vaccine is effective in patients with hematological malignancies and might assist clinical decision making. However, larger studies with clinical outcomes are needed.
Author Response
Dear Reviewer,
thank you very much for reviewing our manuscript. We would like to resubmit a revised version In this cover letter we would list all amendments and responses to the points raised and how we have dealt with them in the manuscript.
Reviewer 2
This is an interesting article dealing with a COVID19 vaccine elicited immunological responses in patients with hematological malignancies. It goes beyond measuring antibody responses and looks at T cell responses.
Some suggestions to the authors:
Point 1: Line 72: This statement is not accurate. The references cited report cohorts of cancer patients with symptomatic COVID 19, and found that risk factors for COVID19 mortality were similar to those of the general population. Except for patients with hematological malignancies, there is no clear evidence that cancer, in itself, is a risk factor for sever COVID19.
Response: For clarity, the sentence reads now as follows:
„Published data indicate that mortality from SARS-CoV-2 infections in patients with cancer is mainly associated with general risk factors such as older age and comorbidities [1,2]. Patients with hematologic malignancies have a particular high risk of Covid-19 related death with ~40% mortality rate [3,4].“
Point 2: Line 126: Were cases and controls matched for age and gender ?
Response: Per protocol, 5- and 10-years frequency-matching in addition to age as a continuous variable were used in the regression models to test the association of age with vaccine-induced humoral and cellular responses. Gender was included as a baseline variable in the statistical analysis as described in point 2.6. „Statistical Analysis“.
For clarity, the following sentence has been included at the beginning of point 3.4. „Predictors of spike-specific immune responses“ to give readers a complete overview of all parameters tested:
Both patient- and vaccine-related factors were evaluated for a potential impact on humoral and cellular immune responses. Patient-related variables included age [continuous variable, 5- and 10-years frequency-matching], gender, baseline B- and T-cell counts, pre-vaccination anti-spike-IgGs, cohort category [controls vs patients], and diagnosis [myeloid neoplasm vs lymphoid neoplasm vs solid tumor receiving PD-L1 inhibition]. Vaccine-related factors were the type of vaccine, number of injections (one versus two), and interval between injections [continuous variable; interval ≤ 35 vs >35 days].“
Point 3: Line 186: what do you mean by “sample size was calculated based on the literature”? sample size should be based on power calculations, which should be based on the estimated difference between patients and controls, based on the literature. Please re-phrase.
Response: The sentence has been re-phrased as follows:
“Sample size was calculated based on published data to the immune response after 30 μg BNT162b2 (Comirnaty ©Biontech/Pfizer) vaccine.[37] Assuming a standard deviation of 0.9 for the logarithm of geometric mean concentrations, enrollment of 236 and 118 evaluable patients and controls respectively would provide 80% power (alpha error, 5%) to detect a significant difference in d35 seroconversions between patients and controls.“
Point 4: Line 218: What were the reasons for not administering the 2nd dose of the vaccine? Did you include the participants that received only the 1st dose in the same analysis as those that received 2 doses ?
Response: In total, 48 study participants (patients n= 22, controls n=26) were administered only one injection. Reasons were: vaccination with the vector-based COVID-19 Vaccine Janssen by ©Johnson&Johnson (n=21), pre-vaccination Covid-19 infection (n=17), and others (n=10).
This is now included in the manuscript under point 3.1. „Patient characteristics“ as follows:
„Reasons for only one injection were vaccination with the vector-based COVID-19 Vaccine Janssen by ©Johnson&Johnson (n=21), a history of a SARS-CoV2 infection prior to vaccination (n=17), and others (n=10).“
As mentioned under Point 2, the type of vaccine and the number of injections (one versus two) were included as vaccine-related variables for both humoral (day 35 and day 120) as well as cellular responses. Interestingly, neither the type of vaccine nor the number of injections were associated with the immune responses. The listing of all patient- and vaccine-related variables at the beginning of point 3.4. „Predictors of spike-specific immune responses“ will simplify the interpretation of the data of point 3.4 without the need for a repeated referral to point 2.6. „Statistical Analysis“ where these variable are mentioned.
Point 5: Line 257: The sentence “with a mean difference … participants” is unclear.
Response: The sentence is now re-phrased as follows:
„However, mean values on d120 were significantly higher across all study participants compared to d35 with a mean difference of 477 U/ml (95%CI, 92.4 - 861.6).“
Point 6: Figure 2: Please add the number of patients represented in each of the graphs.
Response: The number is added in the heading of each paragraph in Figure 2. Additionally, all numbers and median values for humoral and cellular immune responses are demonstrated in detail in table 3.
Point 7: Line 264: It is plausible that among patients with low IgG titers on d35 there would be a hogh percentage of participants that got only the 1st vaccine dose. Please clarify if there was an association between low IgG titers on D35 and receipt of the 2nd vaccine dose more than 35 days after the 1st dose (after testing). Some discussion on this question exists in the “discussion” section (line 362). I would suggest you clarify this discussion and move it to the “results” section.
Response:
As mentioned under Point 2, the interval between injections [continuous variable; interval ≤ 35 vs >35 days] was also a vaccine-related variable that was evaluated for a potential impact on humoral and cellular immune responses. With the already mentioned listing under point 3.4. „Predictors of spike-specific immune responses“, the interpretation of the data becomes easier.
As suggested, the following sentence has been deleted from the discussion and moved to the results section: No significant association of interval between injections with vaccine-induced responses was detected.
Thus, the impact of the interval under point 3.4 reads now as follows:
The interval between injections had no significant impact on vaccine-induced humoral or cellular responses. Although statistically not significant (p=0.05), an interval >35 days between injections tended to be associated with lower d35 but not d120 humoral responses in the lymphoid cohort only.
Point 8: Discussion – the discussion needs editing, both content-wise and language editing. It seems as if it was written by a different author then the rest of the manuscript. I suggest you re- write it and then have a linguistic editor make final corrections. The discussion section should include a paragraph summarizing the results and their novelty, a paragraph on the potential implication of your findings, a paragraph relating to other studies on immunogenicity of COVID vaccines in cancer patients and how your results compare to these studies, and a paragraph on the limitations of your study.
Response: The discussion has been edited as suggested and checked by a native English-speaking colleague. It reads now as follows:
Despite the older age of patients compared to controls, sustainable and/or improvements in seroconversion rates and anti-splike-IgG concentrations over time were observed across all cohorts. As expected, d35 seroconversion was higher in controls (98%) compared to patients (68%) (p<0.001). However, d120 seroconversion improved across all patient cohorts [oncologic malignancies under PD-L1 inhibitor therapies (81% to 100%), myeloid neoplasms (82% to 97%), lymphoid neoplasms (48% to 66%)]. Indeed, patients with myeloid and oncologic malignancies under PD-L1 inhibitor therapies had comparable seroconversion rates to the control group. The few cases with pre-vaccination anti-spike-IgGs without a known history of Covid-19 infection might represent asymptomatic infections or cross-reactive antibodies generated during previous infections with other coronaviral strains [50].
Another key finding was the remarkable and largely seroconversion-independent d120 SARS-CoV-2–specific CD4+TNFa+IFNγ+IL-2+- and CD8+TNFa+IFNγ+IL-2+-cells across all cohorts, particularly in the lymphoid group with the lowest seroconversion rate. Indeed, the cellular response in patients with lymphoid neoplasms with detectable CovCD4+- in 71% and CovCD8+-cells in 31% of cases was comparable to that measured in the control group (CovCD4+- 74% and CovCD8+-cells 42%). A sizeable fraction of lymphoid patients demonstrated T-cell responses without detectable spike-specific-IgGs. Overall, CD4+ T-cell responses outnumbered CD8+ responses in our study. This is in line with what has been observed in immunocompetent individuals [34,51].
To our knowledge, this work is the first to describe the kinetics of SARS-CoV-2 vaccine-induced humoral and cellular responses over time. The impaired early (d35) seroconversion in patients with hematologic malignancies is in line with previous publications [11-26]. However, the majority of patients demonstrate sustained and/or improved humoral and/or cellular responses if measured later (d120). These immune responses were seen irrespective of the type of vaccine or interval between injections. In a recent longitudinal study, antibodies against the SARS-CoV-2 spike antigen and specific memory cell responses were detected in 96% and 63% of health care workers four and eight months post infection [52,53].
In line with the literature, our data imply that the previously reported „early“ T-cell responses [15,24,34-41] are likely to persist for several months in patients with hematologic malignancies after vaccination similar to what has been observed in immunocompetent individuals after COVID-19 infections [53-56].
Taken together, our results underscore the need for large-scale follow-up data to establish standardized post-vaccination time-windows for humoral and cellular response assessments to identify „true vaccination failures“ in cancer patients.
However, the routine applicability of tests to measure humoral and cellular immune responses remains challenging. Although several assays for anti-SARS-CoV-2-IgG are commercially available, current assays generate discrepant results. In fact, we are still far from the identification of optimal thresholds for IgG-positivity as a surrogate for neutralization capacity and neutralizing antibodies (NAbs) which confer protection [29,57-60]. Further, correlations between NAbs and clinical efficacy against infections are weak and likely rely on the population tested [61,62].
The issue is even more complicated regarding cellular response assays. Generally, they are not readily available and mainly used for research purpose. There is often a preponderance of using IFNγ–related readouts to assess T-cell responses [24,63]. However, data suggest that polyfunctional T-cells have higher protective efficacy after vaccination compared to IFNγ monofunctional T-cells [64]. The true percentages of patients developing polyfunctional vaccine-induced CD4+TNFa+IFNγ+IL-2+- and CD8+TNFa+IFNγ+IL-2+-cells might be incorrectly assessed if IL-2 is not considered [43].
Although cellular responses are promising indicators of immunity, our data do not suggest that those with a response compared to those without such a response are more likely to be protected. Yet, even if infections cannot be prevented, it is still possible that T-cell responses are sufficient to ensure mild courses of COVID-19 disease. Thus, studies are necessary to evaluate the degree of cellular-induced clinical protection. Further, SARS-CoV-2 variants such as Omicron (B.1.1.529) with their antibody escape highlight the importance of addressing whether T-cell recognition is also affected.
One limitation is that we did not measure NAbs with virus neutralization assays which are considered to be the gold standard. Yet, we used an anti-SARS-CoV-2 IgG assay with cutoffs for reasonable prediction of NAb [65]. After enrollment started, health authorities in some federal states in Germany changed the interval between injections from 21 to 42 days. The potential impact on d35 response evaluation was discussed with the statistician and accounted for by including the interval as a vaccine-related variable in the regression model. As ~50% of participants across all cohorts received the second injection 42 days later, comparison between groups was feasible. Finally, despite the relatively large number of participants, data of secondary outcomes remain explorative and need to be confirmed in larger trials.
In summary, our longitudinal study describes the nature of SARS-CoV-2 vaccine-induced humoral and cellular immune landscape in patients with hematologic on oncologic malignancies under PD-L1 inhibition.
Point 9: Conclusions: “The findings have implications on clinical decision-making and public health policy for designing vaccine strategies given the current timing and recommendations for SARS- 379 CoV-2 booster doses.” – this seems like an over- statement, since this study did not look at clinical results (infections/ complications of COVID19) and it’s cohorts comprise of 50-100 patients. I would say that the findings suggest that the vaccine is effective in patients with hematological malignancies and might assist clinical decision making. However, larger studies with clinical outcomes are needed.
Response: the wording „and public health policy“ has been deleted and the following sentence included:
„Larger studies with clinical outcomes are needed.“
We hope that we have adequately answered the points raised by the reviewer
Sincerely
Yours
Susann Schulze and coauthors